# Insights into the Methodological, Biotic and Abiotic Factors Influencing the Characterization of Xylem-Inhabiting Microbial Communities of Olive Trees

**DOI:** 10.3390/plants12040912

**Published:** 2023-02-17

**Authors:** Manuel Anguita-Maeso, Juan A. Navas-Cortés, Blanca B. Landa

**Affiliations:** Department of Crop Protection, Institute for Sustainable Agriculture, Spanish National Research Council (CSIC), 14004 Córdoba, Spain

**Keywords:** microbiome, endophytes, xylem, olive, determining factors

## Abstract

Vascular pathogens are the causal agents of some of the most devastating plant diseases in the world, which can cause, under specific conditions, the destruction of entire crops. These plant pathogens activate a range of physiological and immune reactions in the host plant following infection, which may trigger the proliferation of a specific microbiome to combat them by, among others, inhibiting their growth and/or competing for space. Nowadays, it has been demonstrated that the plant microbiome can be modified by transplanting specific members of the microbiome, with exciting results for the control of plant diseases. However, its practical application in agriculture for the control of vascular plant pathogens is hampered by the limited knowledge of the plant endosphere, and, in particular, of the xylem niche. In this review, we present a comprehensive overview of how research on the plant microbiome has evolved during the last decades to unravel the factors and complex interactions that affect the associated microbial communities and their surrounding environment, focusing on the microbial communities inhabiting the xylem vessels of olive trees (Olea europaea subsp. europaea), the most ancient and important woody crop in the Mediterranean Basin. For that purpose, we have highlighted the role of xylem composition and its associated microorganisms in plants by describing the methodological approaches explored to study xylem microbiota, starting from the methods used to extract xylem microbial communities to their assessment by culture-dependent and next-generation sequencing approaches. Additionally, we have categorized some of the key biotic and abiotic factors, such as the host plant niche and genotype, the environment and the infection with vascular pathogens, that can be potential determinants to critically affect olive physiology and health status in a holobiont context (host and its associated organisms). Finally, we have outlined future directions and challenges for xylem microbiome studies based on the recent advances in molecular biology, focusing on metagenomics and culturomics, and bioinformatics network analysis. A better understanding of the xylem olive microbiome will contribute to facilitate the exploration and selection of specific keystone microorganisms that can live in close association with olives under a range of environmental/agronomic conditions. These microorganisms could be ideal targets for the design of microbial consortia that can be applied by endotherapy treatments to prevent or control diseases caused by vascular pathogens or modify the physiology and growth of olive trees.

## 1. Introduction

Plant microbiome research has been expanding over the last years due to its key determinant role in plant health and crop productivity [1], as is revealed by the increase in the number of scientific publications listed in Web of Science (WoS) and Scopus databases (Figure 1). Nowadays, it is generally accepted that plants live in association with a rich diversity of microorganisms. The sum of these microbial cells formed by multi-kingdom microbial communities (bacteria, fungi, viruses, etc.) that colonize both below- and aboveground plant organs in a particular environment is called plant-associated microbiota and refers to the taxonomy and abundance of microbial community members in a given environment. On the other hand, the totality of genomes of this microbiota is called the plant microbiome and it is often used to define the microbial behaviors or functions determined by a microbiota [2].

A better knowledge of the plant microbiome will result in developing alternatives to solve some of the environmental issues and challenges worldwide caused by excess use of chemicals for pest control by enhancing crop production and soil health [3] and by developing biological control alternatives for managing plant diseases [4,5]. Plant-associated beneficial microbiomes can confer fitness advantages to the host plant because these microorganisms can directly impact plant growth and health by producing phytohormones, improving nutrient acquisition and phosphate solubilization and tolerance to biotic and abiotic stresses. Additionally, they can act indirectly by activating a series of host physiological and immune responses that may trigger the multiplication of a specific microbiome to cope with the pathogen infection. This beneficial elicited microbiome can compete for plant resources and niche space or inhibit the growth of plant pathogens via the production of antibiotics, fungal cell wall-degrading enzymes or siderophores, among others [6,7,8].

In this review, we have explored how research on the plant microbiome has evolved during the last decades to unravel the factors and complex interactions that affect the associated microbial communities and their surrounding environment, focusing on a very specific plant niche, the xylem vessels of the olive tree. We have also discussed the importance of studying the xylem-inhabiting microorganisms and their potential role in protecting plants against vascular pathogens, such as *Verticillium dahliae* and *Xylella fastidiosa*, that threaten olive production worldwide, highlighting the current challenges and future perspectives that endophytes represent for the control of these xylem-associated pathogens. Additionally, we reviewed some of the most important biotic and abiotic factors that can modulate and influence xylem microbial communities in different host plants (Table 1).

## 2. Olive Tree and Its Vascular Pathogens

Olive (*Olea europaea* susbp. *europaea* L.) is one of the most relevant tree species within the Mediterranean Basin due to its cultural and economic importance, without underestimating the noteworthy environmental attributes related to its ability to survive in poor, shallow and dry soils, and its contribution to the control of soil erosion and increase in soil nutrient retention [9,10]. In addition, olive orchards are important agroforestry habitats, playing essential roles in maintaining ecosystem diversity [11]. Hence, olive cultivation comprises relevant economic, environmental and socio-cultural values representing an agricultural system that should remain integral to the Mediterranean Basin. However, the health status of olive trees is being threatened by a remarkable increase in diseases caused by xylem-inhabiting pathogens, such as the plant pathogenic bacterium *Xylella fastidiosa* and the soilborne fungus *Verticillium dahliae.* Both pathogens are global threats to olive production which can adversely affect olive growth and production, causing substantial economic losses and severe environmental impact [12,13,14,15,16].

The life cycle of *V. dahliae*, the causal agent of Verticillium wilt, is mainly characterized by its growth confined within the xylem vessels during the pathogenic phase and its ability to survive for many years as dormant microsclerotia in soil or within plant debris [12,17]. *Verticillium dahliae* infects the olive plants through the root system; then, the pathogen colonizes the xylem vessels and impairs the sap flow through mycelial proliferation, the formation of occlusions and tyloses that ultimately may cause the death of the tree [12,18]. The implementation of an integrated management strategy is the best option for controlling this disease, combining the use of resistant olive cultivars, or tolerant ones grafted onto resistant rootstocks, with adequate irrigation management and agricultural practices that prevent the spread of inoculum of the pathogen [12,19,20,21,22,23].

On the other hand, *X. fastidiosa* is a quarantine plant pathogenic bacterium in the European Union (EU) and was ranked first on the list of the 20 priority pests within the EU territory (Commission Delegated Regulation (EU) 2019/1702). *X. fastidiosa* is transmitted by sap-feeding insect vectors and has been described as the causal agent of Olive Quick Decline Syndrome (OQDS). This disease is characterized by rapid dieback of shoots, twigs and branches, followed by the death of the tree [24]. Sap-feeding insect vectors inoculate the pathogen into xylem vessels from infected to healthy trees. Once inside, the aggregation of bacterial cells and biofilm formation block the sap flow in the xylem causing the plant to dry out and die [25]. *Xylella fastidiosa* is taxonomically divided into three major subspecies (*fastidiosa*, *multiplex* and *pauca*) [26], although two additional subspecies have been proposed (*morus* and *sandyi*). Only isolates belonging to the subspecies *pauca* are currently considered a threat to olive production worldwide. As previously indicated for *V. dahliae*, currently, there is no efficient control measure for the management of *X. fastidiosa* once the plant is infected. Consequently, the use of preventive strategies focused on the early detection and eradication of infected host plants, the control of the vectors and restrictions on plant material movements are currently the only available tools for the management of the diseases caused by this harmful bacterium [27,28,29].

The establishment of a pathogen lifestyle in a microorganism depends on its interactions with the host microbiome and the host immune system [30]. The xylem-associated microorganisms are involved in several biotic and abiotic processes within the plant host, including the acquisition of nutrients and increase of plant tolerance to abiotic stresses, without overlooking their role in the defense of the plant against vascular pathogens. In this context, acquiring and maintaining a beneficial xylem-associated microbiota capable of adapting more rapidly to a changing environment could be a selective advantage for the olive tree to fight against vascular pathogens [31,32].

## 3. The Role of Xylem Composition and Its Associated Microorganisms in Plants

Xylem vessels play a decisive role in plant growth maintenance, providing a main route and refined plumbing system for the circulation of micro- and macronutrients derived from xylem sap. This nutrient flow in the xylem from roots to shoots ensures a controlled exchange of ions and metabolites from the xylem to a required and localized area which is driven by hydrostatic pressure and water potential [33,34]. The xylem is traditionally considered a nutritionally poor environment with highly fluctuating negative pressure and low oxygen and nutrient content compared with other plant ecological niches [35]. Consequently, the characterization of xylem sap composition is crucial to understand the nutrient fluxes and dynamics occurring within this plant niche since it may influence the maintenance or selection of a specific beneficial xylem-associated microbiota [36].

Several studies have shown that xylem sap contains a wide variety of chemical compounds, including a high level of sugars, followed by alcohols, amino acids and organic acids [37,38,39]. Xylem sap metabolome analysis demonstrated the presence of these compounds in different crops, including herbaceous (i.e., tomato [40], soybean [38] and cabbage [37]) and woody plant species (i.e., grapevines [41], peach [42] and plum [43]). In olive xylem sap, several primary (mannitol, ethanol, glutamine and acetic acid) and secondary metabolites (terpenoids, phytohormones, alkaloids, sterols, retinols, tocopherols and carotenoids) [44] have been identified [39], many of which have been shown to play a determinant role in plant growth and resistance or tolerance to different biotic and abiotic stresses [45,46]. Additionally, several studies recently showed that olive xylem sap composition can be influenced by a variety of factors, such as changes in crop management [47], climatic conditions and biotic factors including the olive plant age and genotype [39].

Xylem vessels are considered ideal niches for microbial endophytes characterized by specific abiotic and biotic conditions supporting the colonization by endophytes and providing an effective pathway for its distribution throughout the plant and a continuous source of nutrients from xylem sap [48]. Nevertheless, only a few microbes have adapted to exclusively inhabit the xylem (e.g., *X. fastidiosa*), while various pathogens may colonize other plant niches without causing symptoms until they reach the xylem vessels (e.g., *V. dahliae*). Thus, once established in the xylem, these pathogens modulate their physicochemical composition to enhance their growth and virulence [49]. On the other hand, there are microbial endophytes that colonize xylem vessels and plant host tissues internally, evoking a series of critical responses that have beneficial effects on plant growth and may confer protection against vascular plant pathogens [50].

Potential engineering of the plant microbiome is generating exciting results for plant disease control by transplanting specific members of the plant microbiome. However, its practical application in agriculture for the control of vascular plant pathogens is hampered by the limited knowledge of the xylem niche. Knowledge of xylem sap composition may be crucial for improving the isolation and culturability of specific beneficial members of the xylem microbiome to exploit them as biocontrol agents [51,52].

## 4. Methodological Approaches to Study the Xylem Microbiota

### 4.1. Methods Used to Extract Xylem Microbial Communities

The xylem can be considered a hostile environment when compared to other ecological niches of the plant due to its limited nutritional content and specific abiotic conditions (i.e., low oxygen content, negative pressure, etc.). This could explain why only a limited number of microbes and plant pathogens are able to thrive in this environment [35]. Furthermore, only a few studies have identified the diversity of xylem-inhabiting microorganisms in promoting plant health and crop productivity [52,53,54,55,56]. The lack of comprehensive studies may be due, at least in part, to the technical difficulties involved in the isolation of xylem-inhabiting microorganisms.

Alexou and Peuke (2013) [57] described four different methods for collecting the plant xylem sap: (i) root pressure exudate, (ii) Scholander–Hammel pressure vessel, (iii) the root pressurizing method according to Passioura (1980) [58] and (iv) (hand/battery) vacuum pump. Although each method presented advantages and disadvantages, all bring in common the small amount of xylem sap recovered, which might be a limitation for obtaining a representative sample of the microbial communities inhabiting this plant niche.

These techniques have evolved and the possibility of using an external port (40 to 60 cm long) with the Scholander chamber device and the admission of pressure has allowed the extraction of higher volumes of xylem sap from larger and more representative plant portions [39,52,59,60]. However, the extraction of xylem sap using this device is not always feasible due to its high cost for laboratories, in particular when the sampling sites are located at long distances from laboratory facilities, or a large number of samples need to be assessed in a short period. Consequently, other methods for extracting xylem exudates need to be explored to overcome these limitations.

One of the simplest approaches for collecting xylem exudates is to make a cross-section excision in the stem and harvest the sap oozing out through capillary movement by gravity and root pressure [40,61]. Additionally, a portable simple method has been developed for the rapid extraction of xylem sap from stems and petioles using negative pressure generated from handheld needleless syringes [62]. However, these methods do not work for most woody plants due to their high negative stem water potential [63].

Another method used for xylem sap extraction consists of the use of macerated xylem tissue obtained by scraping the most external layers of debarked woody pieces with a sterile scalpel. This method is the most widely used approach in studies targeting the xylem microbiome, since this technique is less time-consuming and more affordable (no need to use a nitrogen tank) than the use of the Scholander device [5,52,60,64]. However, it has been demonstrated that the Scholander chamber device or the woody chips maceration approach retrieve different microbial communities’ profiles, since they affect the recovery of several microbial taxa, showing the importance of the choice of a proper xylem extraction method for the characterization of the xylem microbiome [52,60]. Differences between these two methods might be explained by the extraction of some microorganisms that may occupy intercellular tissues surrounding the xylem vessels that could not be true inhabitants of the vascular tissues, or by the extraction of microbes forming biofilms and strongly adhered to the xylem vessels, the release of which is favored by the tissue maceration process [5].

### 4.2. Assessment of Microbial Communities by Culture-Dependent Approaches

One of the most conventional approaches for the study of plant-associated microorganisms is the use of culture-dependent techniques. Although only a minor fraction of individual microbial species out of the whole microbial community is known to be cultivable, this technique offers several advantages due to the establishment of microbial cultures under laboratory conditions and the purification of single strains from a microbial cluster present in particular ecological niches [65]. This approach has led to the recovery and isolation of a wide range of microorganisms from the rhizosphere [66,67], endosphere [52,68,69] and phyllosphere [70,71] plant compartments.

The development of studies focusing on the optimization of the isolation of specific members of the plant microbiome is rising, as there is a need to test hypotheses based on the manipulation of the plant microbiota and to establish synthetic communities for plant microbiome studies. This strategy can be applied to identify key microbial genera associated with particular plant phenotypes (e.g., resistant to vascular plant pathogens [5]) or by transplanting beneficial bacteria from microbiomes of wild plant genotypes to cropped cultivars [72]. Such cultured bacterial consortia can also be ideal targets for the design of effective biofertilizers, biostimulants or biocontrol agents for a wide range of crops [73]. Therefore, the rebirth of microbial culture collections from specific plant ecological niches represents a valuable tool for increasing our understanding of the plant microbiome. In fact, some authors have applied systematic bacterial isolation approaches to establish culture collections of the *Arabidopsis thaliana* leaf- and root-associated microbiota, being able to capture most of the species found in their respective natural communities (≥0.1% relative abundance) [74].

In line with this, Anguita-Maeso et al. (2020) [52] recovered a bacterial culture collection from the xylem sap of different cultivated and wild olive genotypes, using two of the xylem extraction methods referred to above (Scholander chamber device and woody chips maceration). Results from this study indicated that bacteria from the phyla Actinobacteria, Firmicutes, Bacteroidetes and Proteobacteria clustering into 34 genera could be readily isolated from different olive genotypes. These results agree with the gross taxonomic bacterial distribution of isolates already identified in the olive rhizosphere and phyllosphere [75,76], as well as with the composition of bacterial endophytes identified in other woody plants, such as citrus, grapevine or poplar [77,78,79,80].

### 4.3. Assessment of Microbial Communities by Next-Generation Sequencing

In recent years, improvements in high-throughput sequencing technologies have vastly shifted our understanding regarding the complexity of plant-associated microbiota and has greatly expanded the catalogue of microorganisms known to inhabit plants and their surrounding environment, increasing the identification of millers of uncultivable microorganisms. The use of NGS technologies has opened new opportunities for high-resolution, cost-affordable studies and represents a holistic approach to deepen into the complex biological systems of plant microbiomes [81].

The era of culture-independent plant microbiome study began with an exploratory phase of the diversity and composition of microbial taxa with amplicon-based community profiling approaches. These studies provided insights into the main environmental and biological factors responsible for shaping plant-associated microbial communities [82]. In NGS analysis, the most frequent practice is the use of metabarcoding amplification libraries obtained by amplification from whole isolated DNA of the ITS for fungal communities [83], whereas the 16S rRNA is used for bacteria [84].

The accurate identification of xylem-inhabiting microorganisms could be a determinant for plant protection against vascular pathogens. This accurate identification involves the avoidance of different bias sources in NGS analysis, ranging from the selection of a convenient DNA extraction kit to the appropriate use of specific PCR primer pairs.

The assessment and optimization of a wide range of commercial and ready-to-use DNA extraction kits should be one of the most important initial steps when developing a protocol for plant microbiota analysis due to its potential influence on the structure and diversity of the recovered community profile [85]. Consequently, the bias associated with different commercial kits for DNA extraction in NGS studies must be considered when analyzing the plant-associated microbiome [86,87,88]. Thus, the quality and concentration of DNA obtained, the short processing time and the kit market price must be factors to consider when selecting a suitable DNA extraction protocol. Thus, it has been demonstrated that the method used for DNA extraction can lead to dramatic differences in microbial output composition [86,88], which makes essential the validation of DNA extraction methods with a mock microbial community to ensure an accurate representation of the microbial communities in the samples under study. In fact, the lack of standardization procedures across plant microbiota studies makes comparing them difficult [89]. Several studies have shown the effects of using different commercial kits for DNA extraction of plant-associated microorganisms in NGS studies [59,90,91]. In olive, the comparison of 12 DNA extraction kits indicated that the most accurate description of a bacterial mock community artificially inoculated on xylem sap samples was generated when using the PowerPlant DNA extraction kit (QIAGEN) [59]. The DNeasy PowerSoil kit (QIAGEN) also showed good characteristics and it could be used to analyse olive microbiota, especially in studies in which other plant niches such as the rhizosphere need to be explored. In fact, these kits are recommended for extracting microbial DNA from environmental samples in the Earth Microbiome and the Human Microbiome Projects, two of the largest microbiome initiatives worldwide [92].

Another procedure that may influence the assessment of microbial communities is the use of appropriate PCR primer pairs to avoid undesired co-amplification of mitochondria and chloroplasts, which may strongly affect the characterization of the microbial community composition of plant-associated material [60,93,94,95,96]. Additionally, the results of bacterial profiling can vary considerably according to the hypervariable region that is amplified within the 16S rRNA, which depends on the choice of primers [60,97]. Multiple primer pairs are available for 16S rRNA analysis; consequently, each pair needs to be carefully selected and tested on specific samples to avoid taxonomy biases in the microbial-community analysis, and to allow for a comparison with different datasets [60,98]. For olive microbiome research, the primer pair 799F-1115R provided a higher depth and taxa coverage, with a low percentage of plant mitochondria sequences co-amplified and with the benefit of not needing an additional purification step (i.e., the excision of the PCR-amplification products from agarose gels) as shown for other primer pairs [60].

## 5. Biotic and Abiotic Factors That Influence Xylem Microbiota

While there are a huge number of studies that focus on the description of the soil- and root-associated microbiome, only a few publications have focused on identifying the diversity in the xylem microbiome and its role in plant health and productivity [52,53,54,55,59,99,100]. The lack of comprehensive studies on this plant niche may be due, in part, to the technical difficulties that are involved in the isolation of the xylem-inhabiting microorganisms, as stated above, or due to the microbiome complexity itself, which can also be affected by several biotic (plant age and developmental stage and the host genotype) [39,52,60] and abiotic (agronomic factors, climate, seasonality, etc.) factors [5,39,101,102,103].

### 5.1. Plant-Associated Factors

Several plant-associated factors can modulate and influence microbial diversity in host plants, such as plant niche, plant age or developmental stage, plant genotype and the use of rootstocks (Figure 2). Soil is considered a large reservoir of microorganisms that interact with plants, which are influenced by root exudates and secretions, being crucial for plant growth and health. Interestingly, microorganisms associated with plant roots might help plants access particular chemical pools, such as nitrate, ammonium or amino-acid sources of nitrogen, thereby generating a specificity response in plant–microbe–soil interactions [104]. The belowground fraction of the plant compartment formed by soil, rhizosphere and roots has been widely analyzed over the last years [1,105,106,107,108,109], as well as the aboveground portion composed of stems, fruits and the phyllosphere [110,111,112,113]. Differences in microbial taxa have been found, mainly according to plant niche, which may be explained by the classical definition that each microbial community occupies and colonizes its own plant niche. This niche adaptation may also have a major role in the selective filtering and recruitment of different microorganisms that results in successful colonizers inhabiting the same host niche [8], as reported for various host plants, such as poplar trees [114], agave and cacti species [115,116], *Arabidopsis thaliana* [117,118] and olive [119].

Additionally, several studies have reported differences in microbiome community composition as a function of plant age or genotype, but mainly in herbaceous species (e.g., mustard, potato, Arabidopsis or soybean) [120,121,122,123], with only a few studies in woody crops (e.g., olive, pine, oak) [39,124,125,126]. In woody crops, the use of rootstocks is the most practical, long-term and economically efficient disease control measure for soilborne plant pathogens that must be also considered as a modulating factor of microbiome assemblages in those crops [127,128,129].

On the other hand, the influence of plant age on the chemical composition of xylem sap has been overlooked, as only a few studies have described this interaction in leaves and roots. Nevertheless, some metabolomic studies have shown the existence of some differentiation of chemical compound dynamics during different plant growth stages in various herbaceous plants [92,130,131,132,133], which may result from different biological activities of the plant host or due to variations in soil properties or environmental conditions [130,131]. Plant age has also been reported to have an undoubted effect on the microbial community structure and assembly. Thus, several studies have indicated differences in the microbial community composition according to plant age, but this has been mainly addressed in herbaceous species (i.e., mustard, potato, Arabidopsis and soybean) [120,121,122,123,134] and much less in woody species [125,126], with only two studies conducted in olive [39,124]. In olive, the metabolomic and ionomic profiles of xylem sap have revealed changes according to plant age, showing significantly higher levels of glucose, fructose, sucrose and mannitol, choline, B and PO43− in adult trees, whereas NO3− and Rb were significantly enriched in olive plantlets [39].

Finally, several studies with domesticated crop species have indicated differences in microbiome community composition between wild accessions and modern cultivated varieties, although, as for other features, mainly for herbaceous species (e.g., barley, bean maize, and rice [135,136,137,138]), and less so in woody crops [99,139,140], but including olive [76,141,142,143]. In the same way, to date, there are few studies addressing how the host genotype influences its microbiome [123,144,145], although some works have shown that microbial community adaptation to host genotypes is stronger at early stages, but the relative importance of host genotype declines with time [144]. Some of the studies focusing on olive have shown the existence of significant differences in population densities and community composition of xylem-limited microorganisms according to the olive genotype [39,52]. However, this should be further explored, including the same olive cultivars growing under different environmental conditions, to be able to differentiate between the effect of the olive genotype and that of the environment [146].

### 5.2. Environmental Factors

Plants and their microbial communities are exposed to environmental factors, such as soil physicochemical properties (pH, salinity, texture, structure, moisture, mineral and soil organic matter content, etc.), climatic conditions (temperature, precipitation, seasonality, etc.) and agronomic and farming practices (use of chemical and biofertilizers, cover crops, etc.) [103]. In this context, a strong seasonality effect on the diversity and composition of the xylem microbiome has been reported for grapes [54] and olives [146] and Anguita-Maeso et al., unpublished results. Furthermore, numerous studies using metabarcoding sequencing have proved that soil type is a major factor driving root microbiome structure [147], while soil pH exerts a stronger impact on total community composition, even at very high taxonomic ranks [148]. Additionally, drought is one of the abiotic factors that more strongly affect plant-associated microorganisms and is positioned as a determinant environmental stressor in agriculture that can alter the composition of root microbiomes by modifying the abundance of certain microbial taxa [106,149]. On the other hand, some common agricultural practices, such as crop rotation, application of chemicals (e.g., ammonium sulphate, ammonium phosphate and urea) and natural fertilizers (e.g., seaweed products and manure) currently used by farmers to improve yield and quality of their products could have a positive, neutral or negative impact on the diversity and structure of plant-associated microbial communities [150].

### 5.3. Infection by Vascular Pathogens

Plants have evolved their own adaptations to alleviate most biotic and abiotic stresses in nature, and they rely on microbial partners to survive and defend themselves against harmful microbial invaders and pathogens [151]. Pathogenic organisms require recognition and communication with a host to suppress their immune response, allowing the pathogen to enter the host tissue and infect it [152].

**Table 1 plants-12-00912-t001:** Summary of main methodological, abiotic and biotic factors that influence the characterization of the xylem-associated microbiome.

Methods for Extracting the Xylem Microbiome	Reference
- Root pressure exudate method - Scholander–Hammel pressure vessel method - Root pressurizing method according to Passioura - Vacuum pump - Scholander chamber with external port - Cross-section excision and capillary movement - Negative pressure with handheld needleless syringes - Macerated xylem tissue	[57] [57] [58] [57] [39,52,59,60] [40,61] [62] [5,52,60,64]
**Methods for isolation and identification of microbiome**	
- Culture-dependent approaches - Culture-independent approaches (NGS)	[52,66,67,68,69,70,71] [85,86,87,88,93,94,95,96]
**Biotic and abiotic factors**	
- Plant-associated factors:	
Ecological niche	[114,115,116,117,118,119]
Plant age	[39,120,121,122,123,124,125,126]
Plant genotype	[39,52,123,144,145]
- Environmental factors:	
Seasonality	[54,146]
Soil type	[147]
Soil pH	[148]
Drought	[106,149]
Agricultural practices	[150]
- Infection by vascular pathogens:	
*Verticillium dahliae*	[15,153]
*Xylella fastidiosa*	[64,154,155,156,157]

Several studies have described differences in plant microbiota when comparing infected and healthy plants, reporting changes in plant host microbiota by direct inhibition or/and by indirectly shifting the microbiome composition in infected plants [5,64,158,159]. In this context, some authors have studied the effect of some vascular pathogens, including *X. fastidiosa* [154,155,156,157] and *V. dahliae* [15,153] in the metabolomic and ionomic profiles of olive trees. Thus, specific chemical compounds detected in stems, leaves and fruits have been indicated to potentially be involved in the defense mechanisms of the host plant against those pathogens. In parallel, some differences in xylem sap microbial composition have been associated with the differential disease reaction observed in certain olive genotypes to those vascular pathogens [102,160]. A recent study described the changes in the diversity and structural profile of the microbial communities associated with the infection of almond trees by *X. fastidiosa* [64], showing that *X. fastidiosa* infection reshapes the xylem-associated microbiome. Finally, some studies have indicated that a modification of the microbial communities inhabiting the xylem vessels, or a reduction of their diversity, can induce changes in the resistance response of the host plant [5]. Moreover, modification of microbial composition during olive propagation, which is conducted in vitro under aseptic conditions, can induce changes and alter the composition of the xylem microbiome by excluding some beneficial endophytes which were determinant in the breakdown of resistance in a wild olive genotype to the vascular plant pathogen *V. dahliae* when grown in infested soil [5].

## 6. Future Perspectives for Xylem Microbiome Analysis

Bioinformatics analyses represent a valuable tool for characterizing the diversity of plant-colonizing microorganisms and their functions and interactions in a particular plant niche and will lead to a better understanding of the plant microbiome for its exploitation in agriculture to improve plant growth, health and crop productivity [119]. Microbial network analysis is a novel exploratory data analysis technique that can be used to derive hypotheses from the microbial composition of thousands of samples sequenced by NGS. Network analysis represents an approach for exploring and identifying patterns in large, complex datasets that are difficult to detect using the standard alpha/beta diversity metrics widely used in microbial ecology and may help to investigate potential associations among microbial taxa across spatial or temporal gradients [161]. Exploring co-occurrence patterns between microorganisms can help to identify potential environmental factors (biotic and abiotic interactions), habitat affinities linked to a variety of inter-dependent factors or shared physiologies among community members that could guide more focused studies or experimental perspectives.

Several authors have described the importance and ecological interpretations of the cross-kingdom co-occurrence networks in the plant microbiome [162,163]. Network analyses have identified the existence of interactions among the myriad of endophytes present in natural communities and other potential higher-order interactions [164]. Thus, this analysis has contributed to the discovery of hub or keystone microbial taxa in the plant microbiome. In fact, leaves of *Arabidopsis thaliana*, a fungal species, *Dioszegia* sp., and an oomycete, *Albugo* sp., were detected as hubs in the networks and experimentally validated as keystone species [165]; whereas in the wild rice seed microbiome, two fungal species were identified to be hub species in the cross-kingdom network [166]. This data analysis tool can be used to determine the existence of aggregation or exclusion interactions in xylem-associated microbiomes and reveal keystone microbial players in the xylem of olive trees to fight against vascular pathogens to contribute to the production of olive plants that are more resilient to these pathogens.

Amplicon profiling approaches have provided insights into community structure and phylogenetic diversity of plant-associated microbes over last decades. However, there is a need to move forward from the era of describing the taxonomic profile of plant niches in order to unravel the microbial activities encoded within the plant holobiont. This step is critical to understanding how microbial communities can execute plant growth and plant protection actions, in order to exploit them as a potential tool to increase plant fitness and health [2]. In this regard, metagenomics, as a cutting-edge technique, is the most recent technology that has emerged to address these questions. The plant-associated metagenomes are defined as the set of genes encoded by any particular microbiota that can provide the host plant with critical nutrients, protection from plant pathogens, production of functional plant hormones and tolerance to abiotic stresses, etc. Metagenomics will also contribute to improving our understanding of species composition, genetic diversity, inter-species interactions and species evolution in the context of different ecological niches [164]. However, to the best of our knowledge, to date, only one study has analyzed the olive xylem microbiome by metagenomic analysis, and was conducted by Giampetruzzi et al. (2020) [102]. These authors studied the dynamics of *X. fastidiosa* infections and its associated endophytic microbiome in susceptible ‘Kalamata’ and resistant ‘FS17’ olive cultivars at two different stages of pathogen infection, concluding that the bacterial and fungal communities present in the xylem of olive trees appeared to be more tightly structured by season and *X. fastidiosa* abundance than by host cultivar, probably due to the high pathogen inoculum pressure in the orchard where olive trees were sampled [102].

The next generation of cultured-based methods, commonly known as culturomics, has been declared the new implement on the block of ‘omics’ approaches. Microbial culturomics has emerged as a successful tool to isolate high numbers of microbes and identify new species that will complement molecular techniques, providing another approach for determining the composition of microbial populations living in diverse environments [167]. This strategy overbears the importance of isolating single cells of environmental microbiomes and bringing them into cultivation. It needs to be accomplished based on the diversification and multiple combinations of various artificial growth media, incubation conditions simulating environmental conditions of the niche of study and the duration of incubation time to favor rare or slow-growing microorganisms. Consequently, there is still a long way to go before standardizing culturomics, which is essential to complement the data obtained from culture-independent NGS analyses [168].

Improving microbial culturability is also essential for a better understanding of plant–microbiota interactions and their functions in the complex ecological niches of the plant holobiont. Culturomics has been designed, optimized and largely applied in human microbiology, but has not yet been fully exploited in plant microbiology [169]. In fact, only a few studies have focused on the design of new media and approaches to improve the culturability of novel bacterial taxa from the uncultured fraction of plant-associated bacterial communities [170,171]. Most of the known microbial phyla (ca. 120 bacterial and 20 archaeal phyla) contain only a few cultivable representatives [172]. Nowadays, no culture media are described for axenic growth of xylem microbiota, which is crucial to understand the ecological interactions of xylem-inhabiting microorganisms with vascular pathogens. For that reason, the characterization of different broth culture media mimicking xylem sap composition for their ability to sustain growth of olive xylem-inhabiting microorganisms is needed to decipher the ecological interactions of xylem-inhabiting pathogens with the indigenous plant endophytic microorganisms. In this regard, we have evaluated four solid media [52] and six broth culture media mimicking the olive xylem sap composition [173] (Anguita-Maeso et al. *unpublished results*) to support the growth of olive xylem-inhabiting microorganisms. A total of 66 genera of olive xylem-inhabiting bacteria were identified that could be cultured in any of the culture media tested, from which, about 42% were previously described as bacterial endophytes of the plant stem in other studies, and the remaining ones are novel cultivable bacteria associated with the plant endosphere. These results can be relevant for future studies aimed at culturing xylem-inhabiting microorganisms potentially involved in host tolerance and/or plant defense against xylem-inhabiting pathogens, or may help to select synthetic microbial communities that are easy to culture and that can be tested *in planta* through endotherapy treatment to determine their ability to suppress vascular diseases.

Thus, the application of xylem microbial consortia is an emerging approach to promote plant growth and biocontrol of vascular plant pathogens. The application of microbial consortia into the xylem vessels by endotherapy can be a powerful tool for modifying the native xylem microbiome of a woody plant to control diseases caused by xylem-inhabiting pathogens or the plant physiology and growth. These microbial consortia can be designed or obtained by two approaches: (1) by extracting the natural xylem microbiome from a different genotype (e.g., a resistant variety) and transplanting it into the desired plant genotype; or (2) it can be produced by in vitro culturing of selected microorganisms independently or as a consortium.

In regard to the first approach, a pilot experiment of a xylem endotherapy treatment was conducted in our laboratory and focused on the modification of the xylem microbiome composition of plantlets of two cultivated olive cultivars (‘Picual’ and ‘Arbequina’) by transplanting an external xylem microbiome obtained from a wild olive genotype, ‘Acebuche’, which showed a fairly different microbial community composition compared to that of the cultivated genotypes. Then, we evaluated the changes in the xylem microbial composition of the challenged plants over time. Results indicated that treated ‘Picual’ and ‘Arbequina’ olive genotypes showed new bacterial species in the first months after the endotherapy treatment. Additionally, although there were no clear differences in the beta-diversity according to the endotherapy treatment, some trends were observed across the different sampling times in each olive genotype that were mainly associated with the plant growth stage, rather than the endotherapy treatment. These results indicate that the xylem microbiome of olive plants is somehow resilient to drastic changes from invading microorganisms [174] (Anguita-Maeso et al., *unpublished results*).

For the second approach, endotherapy treatments with microorganisms artificially cultured prior to inoculation have already been explored to control some plant pathogens of woody trees, including beech [175] and elm [176]. Currently, we are evaluating the possibility of modifying the xylem microbiome composition of olive by transplanting a microbiome artificially cultured in liquid media mimicking the xylem sap composition and monitoring the stability of the challenged microbial community composition over time. When optimized, this technique can settle the basis to inoculate ad hoc selected microbial consortia or synthetic communities (SynCom) to control diseases caused by xylem-inhabiting pathogens, such as *V. dahliae* or *X. fastidiosa,* and/or to modify olive plant physiology and growth.

Finally, the plant-associated microbiome is not only composed of prokaryotic organisms (e.g., bacteria and archaea), but also of other less studied biota members, including unicellular (e.g., protozoa) and multicellular (e.g., fungi) eukaryotes and viruses, which all may play important roles in plant growth and should not be overlooked in future studies [177]. Therefore, additional research into the dynamic interactions between or among these less represented biota communities within the plant microbiome are necessary to better guide the harnessing of the plant microbiome to increase crop yield and quality, as well as to increase its resilience against vascular pathogens. One approach to directly explore the whole plant microbiome is based on an amplification-free NGS approach, defined as shotgun whole genome sequencing. This approach can be technically employed for general plant microbiome identification, but can also be used specifically for the detection of vascular pathogens, paving the way for novel sequencing opportunities as an effective surveillance tool for the detection of plant pathogens [178].

## Figures and Tables

**Figure 1 plants-12-00912-f001:**
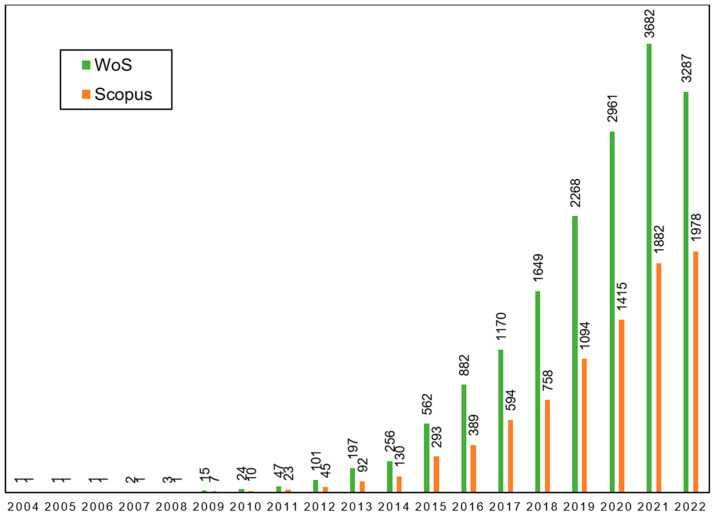
Number of publications including “plant microbiome” in the title and/or as keywords in Web of Science (WoS) and Scopus databases. Review was performed in January 2023.

**Figure 2 plants-12-00912-f002:**
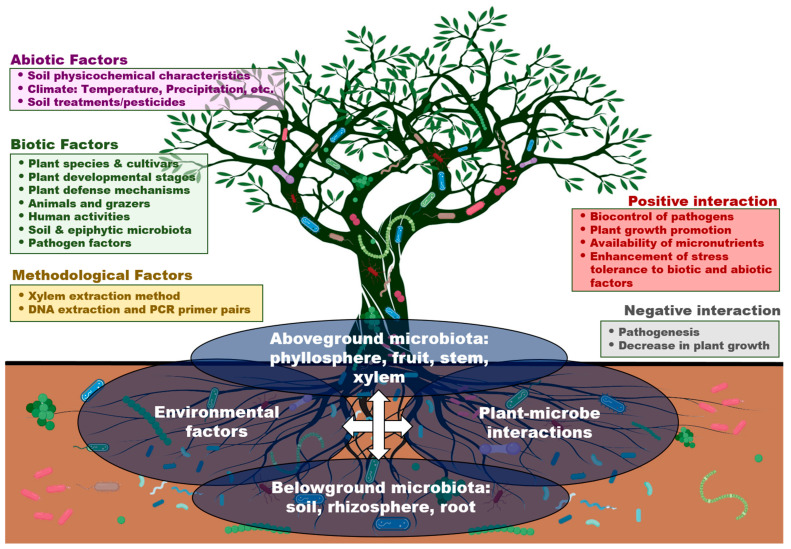
Main methodological, abiotic and biotic factors that influence the characterization of the xylem-associated microbiome and its interactions in olive.

## Data Availability

No new data were created or analyzed in this study. Data sharing is not applicable to this article.

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
