# Peer review of "Insights into the Methodological, Biotic and Abiotic Factors Influencing the Characterization of Xylem-Inhabiting Microbial Communities of Olive Trees"

_plants, 2023, doi:10.3390/plants12040912_

Round 1

Reviewer 1 Report

Review paper entitled “Insights into the methodological, biotic and abiotic factors influencing the characterization of xylem-inhabiting microbial communities of olive trees” contain some information which authors tried to present in this review, below is the comments related with this manuscript.

Title of the review should be more specific to the contents

Firsts paragraph of the abstracts require extensive English improvements

Review contains the required and updated information, additionally text contents large numbered of required references which is related to the topic.

Although I found this as informative related with the topic, but my major concern is related with umber of figures and table, as figure and table improves the attractiveness of information.

Hence I suggest at lists few tables related to the work should be added, additionally more informative figures also required.

Author Response

Review paper entitled “Insights into the methodological, biotic and abiotic factors influencing the characterization of xylem-inhabiting microbial communities of olive trees” contain some information which authors tried to present in this review, below is the comments related with this manuscript.

Title of the review should be more specific to the contents

Thanks for your comment. We pointed out in the tittle all main sections that we cover in the review (i.e, the methodological, biotic and abiotic factors as the main factors that may influence the characterization of xylem-inhabiting microbial communities). If we are more specific to the content of the review, this implies including the details of types of methodologies (extraction method, culture-dependent or NGS approach) or the covered biotic and abiotic factors (plant and environmental factors or the presence of pathogens, etc.). So, we consider that by adding more specific information, the tittle will become too long and it will divert attention from the main question of the review. For that reason, we think it is better to keep it as it is.

Firsts paragraph of the abstracts require extensive English improvements

Thanks for your suggestion. We made English improvements on that paragraph and throughout the entire manuscript.

Review contains the required and updated information, additionally text contents large numbered of required references which is related to the topic.

Thanks for your comment.

Although I found this as informative related with the topic, but my major concern is related with number of figures and table, as figure and table improves the attractiveness of information. Hence I suggest at lists few tables related to the work should be added, additionally more informative figures also required.

Thanks for your suggestion. We included a table that summarises the content of the review in order to highlight the most important topics, so the reader can easily identify the information of interest.

Reviewer 2 Report

The manuscript reviews literature regarding several aspects of classical and molecular microbiology, all tied together by a central topic that is olive trees and their associated xylematic pathogens, with particular mention of Xylella fastidiosa.

Overall, I find the review to be well-organized and comprehensive.

I only have minor comments/suggestions for the authors:

1) I'd like to suggest some more careful consideration of some terms that are used. For example, the sentence in lines 45-46 seems to be talking about the whole microbiota, but uses the term "commensal microorganisms"; Commensal is a term that has several different meanings depending on the specific literature cited, but I do not feel it is the best suited to describe the entirety of plant-associated microorganisms found in the microbiota. Likewise, in lines 58-59 the term "plant-associated microbiomes" is used as the subject of some sentences that describe beneficial effects that the microbiome can have on the host, but the plant-associated microbiome can include pathogens and non-beneficial microorganisms. These are just examples made to highlight what I mean with my suggestion of considering carefully the terms to use in the different sentences.

2) Line 101-102 reports X. fastidiosa as a quarantine plant pathogen in the European Union, but it would be more precise to state that it is one of the 20 organisms that are not only regulated by quarantine but designed as a priority pest, which is an even more stringent regulation than regular quarantine.

3) In lines 107-108 the authors report that X. fastidiosa is divided in 3 subspecies. While I personally agree with this taxonomical categorization, I believe that in a review it should be mentioned that there are several more subspecies that have been proposed and that the debate on the taxonomy of the pathogen is still quite open.

4) There is an apparent contradiction in the information reported by the authors: in lines 149-151 it is stated that the xylem is considered an ideal niche for the development of endophytes, while in lines 167-168 it is stated that the xylem is an hostile environment for microorganisms. While I do understand that both statements are correct, I believe these sentences should be changed to be more harmonious with one another.

5)In line 192 there is a problem of concordance in the sentence "these method does not work".

6) I believe that in lines 210-211 the sentence should be rephrased. In the current form it seems that the sentence refers to a little number of whole communities being cultivable, while I believe the intended meaning is that only a little number of individual bacterial species out of the whole community is cultivable.

7)Line 253: ITS is reported as rRNA: it is an Internal Transcribed Spacer between two rRNAs but it is not an rRNA sequence itself.

8)For the paragraphs spanning lines 255 to 290 I'd suggest to remodulate the information a bit, first presenting the main sources of bias in NGS analysis, and then going in depth into them, rather than going in depth on the bias from extraction protocols without even mentioning the primer bias. Also, I believe that a mention should be made of amplification-free NGS approaches (e.g. whole-metagenome shotgun), either here or in the future perspectives. At the best of my knowledge, an amplification-free approach was used specifically for X. fastidiosa detection in the study by Faino et al., 2021 (doi: 10.1111/ppa.13416), but it could be technically employed also for general microbiome description.

9)As another suggestion to the authors, still in the future perspectives, I'd suggest to add some mentions to the other under-developed studies on viruses and Archea, which are always the least-considered parts of the microbiome but should be less overlooked in future studies.

Author Response

I only have minor comments/suggestions for the authors:

1) I'd like to suggest some more careful consideration of some terms that are used. For example, the sentence in lines 45-46 seems to be talking about the whole microbiota, but uses the term "commensal microorganisms"; Commensal is a term that has several different meanings depending on the specific literature cited, but I do not feel it is the best suited to describe the entirety of plant-associated microorganisms found in the microbiota. Likewise, in lines 58-59 the term "plant-associated microbiomes" is used as the subject of some sentences that describe beneficial effects that the microbiome can have on the host, but the plant-associated microbiome can include pathogens and non-beneficial microorganisms. These are just examples made to highlight what I mean with my suggestion of considering carefully the terms to use in the different sentences.

Thanks for your considerations. That was true, we removed the term “commensal” and we include the word “beneficial” to avoid these misunderstandings.

2) Line 101-102 reports X. fastidiosa as a quarantine plant pathogen in the European Union, but it would be more precise to state that it is one of the 20 organisms that are not only regulated by quarantine but designed as a priority pest, which is an even more stringent regulation than regular quarantine.

We agreed with your statement. We added this information to highlight the importance of X. fastidiosa in the European Union as a priority pest on page 4, third paragraph.

3) In lines 107-108 the authors report that X. fastidiosa is divided in 3 subspecies. While I personally agree with this taxonomical categorization, I believe that in a review it should be mentioned that there are several more subspecies that have been proposed and that the debate on the taxonomy of the pathogen is still quite open.

We added that the Xylella fastidiosa subsp. sandyi and morus have also been proposed.

4) There is an apparent contradiction in the information reported by the authors: in lines 149-151 it is stated that the xylem is considered an ideal niche for the development of endophytes, while in lines 167-168 it is stated that the xylem is an hostile environment for microorganisms. While I do understand that both statements are correct, I believe these sentences should be changed to be more harmonious with one another.

We rephrased both sentences to be more precise and not be contradictory.

5)In line 192 there is a problem of concordance in the sentence "these method does not work".

Sorry, we corrected this grammatical mistake.

6) I believe that in lines 210-211 the sentence should be rephrased. In the current form it seems that the sentence refers to a little number of whole communities being cultivable, while I believe the intended meaning is that only a little number of individual bacterial species out of the whole community is cultivable.

Yes, we agreed with your point. We rephrased this sentence to empathize your statement.

7)Line 253: ITS is reported as rRNA: it is an Internal Transcribed Spacer between two rRNAs but it is not an rRNA sequence itself.

Yes, that is correct. It was a misspelling. Thanks for noticing it.

8)For the paragraphs spanning lines 255 to 290 I'd suggest to remodulate the information a bit, first presenting the main sources of bias in NGS analysis, and then going in depth into them, rather than going in depth on the bias from extraction protocols without even mentioning the primer bias. Also, I believe that a mention should be made of amplification-free NGS approaches (e.g. whole-metagenome shotgun), either here or in the future perspectives. At the best of my knowledge, an amplification-free approach was used specifically for X. fastidiosa detection in the study by Faino et al., 2021 (doi: 10.1111/ppa.13416), but it could be technically employed also for general microbiome description.

Thanks for your suggestions. We added an introductory paragraph before going to depth on the potential biases in NGS approaches. We also included the whole-metagenome shotgun approach in the last section on future perspectives.

9)As another suggestion to the authors, still in the future perspectives, I'd suggest to add some mentions to the other under-developed studies on viruses and Archea, which are always the least-considered parts of the microbiome but should be less overlooked in future studies.

We agreed with you. We added some information regarding viruses and Archaea in future perspectives.

Round 2

Reviewer 1 Report

Authors made substantial changes as per comments and suggetion. Now manuscript can be considered after final English and Grammar check.